# Transcriptome Analysis Reveals the Response Mechanism of *Frl*-Mediated Resistance to *Fusarium oxysporum* f. sp. *radicis-lycopersici* (FORL) Infection in Tomato

**DOI:** 10.3390/ijms23137078

**Published:** 2022-06-25

**Authors:** Yuqing Sun, Huanhuan Yang, Jingfu Li

**Affiliations:** College of Horticulture, Northeast Agricultural University, Harbin 150030, China; sunyuqing0061@163.com

**Keywords:** *Fusarium* crown and root rot, *Fusarium oxysporum* f. sp. *radicis-lycopersici*, *Frl*, RNA-seq, resistance response

## Abstract

Tomato *Fusarium* crown and root rot (FCRR) is an extremely destructive soil-borne disease. To date, studies have shown that only plants with tomato mosaic virus (TMV) resistance exhibit similar resistance to tomato *Fusarium oxysporum* f. sp. *radicis-lycopersici* (FORL) and have identified a single relevant gene, *Frl*, in Peruvian tomato. Due to the relative lack of research on FCRR disease-resistance genes in China and elsewhere, transcriptome data for FORL-resistant (cv. ‘19912’) and FORL-susceptible (cv. ‘Moneymaker’) tomato cultivars were analysed for the first time in this study. The number of differentially expressed genes (DEGs) was higher in Moneymaker than in 19912, and 189 DEGs in the ‘plant–pathogen interaction’ pathway were subjected to GO and KEGG enrichment analyses. *MAPK* and *WRKY* genes were enriched in major metabolic pathways related to plant disease resistance; thus, we focused on these two gene families. In the early stage of tomato infection, the content of JA and SA increased, but the change in JA was more obvious. Fourteen genes were selected for confirmation of their differential expression levels by qRT-PCR. This study provides a series of novel disease resistance resources for tomato breeding and genetic resources for screening and cloning FORL resistance genes.

## 1. Introduction

Tomato *Fusarium* crown and root rot (FCRR), an extremely destructive soil-borne disease that was first identified in 1974, is induced by *Fusarium oxysporum* f. sp. *radicis-lycopersici* (FORL) [1]. In 1976 [2], FORL was discovered in South America. To date, FORL has caused heavy losses in tomato production in various countries, such as Israel, Japan, America, Mexico, Korea, Canada, and parts of Europe, representing one of the major soil-borne diseases in tomato after root-knot nematode infestation and tomato yellow leaf curl virus (TYLCV) [3,4,5,6].

A single gene, *Frl*, from Peruvian tomato was identified in germplasm material with significant resistance to FORL [7], and research has shown that plants with tomato mosaic virus (TMV) resistance exhibit a similar resistance to tomato FORL [8]. However, limited test conditions at that time likely prevented the identification of the specific site for the *Frl* gene to clone and verify its function. Few studies on the tomato FORL resistance gene *Frl* have been conducted in China to date, and the available research on *Frl* from other countries has been limited since the end of the last century.

Plant resistance to pathogens depends on plant native immunity, namely, PAMP-triggered immunity (PTI) and effector-triggered immunity (ETI), which are associated with salicylic acid (SA) and jasmonic acid (JA). As an important signalling molecule, SA exerts a pivotal regulatory effect on plant defence against pathogen invasion [9,10,11,12]. The accumulation of SA in plants greatly increases the expression of pathogenesis-related (PR) genes [13]. Recent studies have shown that several components of the *WRKY* transcription factor (TF) family play an important role in initiating SA biological synthesis [14], and *WRKY* transcription factors are widely believed to play a positive or negative regulatory role in plant defence and SA synthesis [15]. *WRKY70* expression is activated by SA but inhibited by JA, and this gene serves as a node in the regulatory interaction between SA and JA signalling in the process of plant resistance to pathogen invasion [16,17]. One of the major roles of JA in plants is to initiate defence responses and participate in reactions to both biotic and abiotic stress [18]. JA activates the transcription factor *WRKY57* as plant leaves age [19]. Current studies indicate that *CDPK* and *MAPK* affect JA synthesis and core pathway functions. Two kinases in *MAPK* are involved in the regulation of JA in plant defence. *MAPK3–6* destroy the structural stability of *MYC*_2_ by phosphorylation and negatively regulate the JA defence pathway. However, *MAPK3–6* activation depends on the COI1 protein, representing the negative regulatory mechanism of the JA pathway [20,21,22]. Phosphorylation of the *MAPK* wound-induced protein kinase (WIPK), another injury-induced protein kinase, is positively correlated with JA content and the strength of the plant defence response [23].

At present, the mechanism of *Frl*-mediated disease resistance is unclear, and we aimed to further explore the response of disease-related genes. Herein, RNA sequencing (RNA-seq) was employed to determine the causal link between *Frl*-mediated disease tolerance and tomato *Fusarium* crown and root rot (FCRR). The outcomes in the present research are valuable not only for revealing the cause of *Frl*-mediated tolerance to FCRR but also for offering a series of novel disease resistance resources for tomato breeding.

## 2. Results

### 2.1. Inoculation

As shown in Figure 1a,d, no marked differences in the stems were found between MM (Moneymaker) and 19912 (*Frl*) tomatoes before inoculation. At 3 dpi (day post-inoculation), scattered necrosis of plant tissue appeared around the inoculation sites on MM (Figure 1b), whereas no obvious signs were observed on the stems of *Frl* tomato plants (Figure 1e). At 6 dpi, the stems of MM widely developed rot (Figure 1c), and the stems of *Frl* plants showed some significant chlorotic spots, which were regarded as signs of a HR (hypersensitive response) (Figure 1f). These typical symptoms revealed that manual inoculation was successfully completed, and samples were harvested at different time points for subsequent analyses.

### 2.2. Microscopic Analysis of FORL Growth in Two Tomato Cultivars

FORL infections in the *Frl* and MM stems were studied using optical microscopy, scanning electron microscopy, and transmission electron microscopy. As shown in Figure 2a,f, no obvious differences were noted between *Frl* and MM stems at 0 dpi. The spores started to germinate at 1–2 dpi (Figure 2b), and hyphae invaded the stomata in *Frl* and MM tomato stems (Figure 2c). At 3 dpi, mycelia and necrotic spots appeared on MM tomato stems (Figure 2d), and HR was observed on the stems of the *Frl* tomato plant (Figure 2g). At 6 dpi, with the development of the disease, mycelial invasion continued, the diseased area in MM tomato stems continuously expanded, and the chloroplast membrane was destroyed, which causes serious disease in the plant (Figure 2e). However, in resistant cultivar 19912, mycelium growth was limited, and the chloroplast structure was intact (Figure 2h). Plants carrying the *Frl* resistance gene (19912) exhibited evident HR posterior to the FORL exposure site, but susceptible plants (MM) showed continuous mycelium growth.

### 2.3. RNA Sequencing and Identification of Transcripts

In this study, a total of 18 samples (3 biological replicates from each cultivar at 0, 3, and 6 dpi) were sequenced using the GNBseq platform, which produced a mean of 6.65 Gb of data for each specimen (Appendix A). Filtering the obtained reads yielded 44.12–45 million clean reads. The clean read Q20 value was greater than 95%, and the Q30 value was greater than 90% (Appendix A). We then used HISAT to compare the clean reads to the reference genome sequence (https://www.ncbi.nlm.nih.gov/assembly/GCF_000188115.4 (accessed on 1 April 2021)). At least 90.88% of the clean reads matched the reference genomic data, among which over 88.4% were uniquely mapped reads. Ultimately, 672 novel DEGs and 1043 novel transcripts were obtained.

### 2.4. DEGs (Differentially Expressed Genes) in Response to FORL Inoculation

DEGs were selected according to a *p* value ≤ 0.01 and a log_2_ fold change ≥2 in response to FORL inoculation of *Frl* and MM tomato plants (Table 1). From 0 to 3 dpi, more downregulated DEGs than upregulated DEGs were observed in *Frl* and MM samples. From 3 to 6 dpi, fewer downregulated DEGs than upregulated DEGs were observed in *Frl* and MM samples. Throughout the process of plant–pathogen interaction, more downregulated DEGs than upregulated DEGs were noted in both *Frl* and MM samples.

As shown in Figure 3a, at 3 dpi, the number of shared DEGs in the two varieties was 659; at 6 dpi, the number of shared DEGs in the two varieties was 179 (Figure 3b); and between 3 and 6 dpi, the number of shared DEGs in the two tomato varieties was 855 (Figure 3c). These results showed that most of the shared DEGs were found in the two tomato varieties when the inoculation time reached the third day, when the immune mechanism triggered by resistance genes was the most active. When the inoculation time reached the sixth day, the number of shared DEGs between the two tomato varieties was significantly reduced, when the immune mechanism triggered by resistance genes was less active. Therefore, we focused on the mechanism of the resistance response during the period from 0 to 3 dpi.

### 2.5. KEGG Pathway and GO Enrichment Analysis of DEGs

To clarify the function of the DEGs associated with the response to FORL, 189 DEGs in the ‘plant–pathogen interaction’ pathway were subjected to GO analysis and KEGG enrichment analysis [24]. Globally, GO is a standardised system for genetic function categorisation that provides a shared vocabulary to comprehensively delineate the attributes of genes and genetic products. There are three primary ontologies describing molecular functions, cell constituents, and biological processes in GO analysis. Through the analysis of significantly enriched KEGG pathways, the primary biological chemical metabolism channels and signalling pathways involving the candidate genes were identified. As presented in Figure 4a, the channels ‘plant–pathogen interaction’ and ‘*MAPK* signalling pathway-plant’ were remarkably enriched (Q-value ≤ 0.05). In addition, GO terms, including ‘plasma membrane (GO_Component)’ (Figure 4b), ‘calcium ion binding (GO_Function)’ (Figure 4c), ‘protein autophosphorylation (GO_Process)’ (Figure 4d), and 19 other terms (Q ≤ 0.05) were remarkably enriched.

### 2.6. Gene Co-Expression Network Analysis

Weighted gene co-expression network analysis (WGCNA) can rapidly identify genetic co-expression modules associated with specimen features from intricate data for subsequent analyses [25]. Seven diverse modules were acquired via a genetic tree coloured according to the relationship between genetic expression levels (Figure 5a), and among these, there were obvious differences in the expression of four modules between the *Frl* and MM samples. As shown in Figure 5b, expression of the genes in MEblack was high in *Frl* at 0 dpi, but expression of the genes in MEcyan was high in MM at 0 dpi. After infection, expression of the genes in MEpink and MElightcyan was high in *Frl* and MM samples at 3 dpi, respectively. Interestingly, expression of the genes in MEblue was high in the *Frl* sample at 6 dpi. Our team performed KEGG analysis on the genes in these three modules. In MElightcyan (Figure 6a), pathways related to ‘ribosome’, ‘porphyrin and chlorophyll metabolism’, ‘fatty acid biosynthesis’, and ‘photosynthesis’ were enriched. In the MEpink module (Figure 6b), pathways associated with ‘RNA transport’, ‘ribosome biogenesis in eukaryotes’, ‘oxidative phosphorylation’, ‘mRNA surveillance pathway’, and the ‘citrate cycle’ were enriched. In the MEblue module (Figure 6c), ‘sulphur metabolism’, ‘cysteine and methionine metabolism’, ‘microbial metabolism in diverse environments’, and ‘plant–pathogen interaction’ pathways were enriched.

### 2.7. Validation of DEG Expression Patterns

To verify the accuracy of the DEG expression patterns indicated by the RNA sequencing results, we analysed 14 DEGs by qRT-PCR with three biological replicates. These 14 genes were involved in KEGG pathways that displayed remarkable enrichment: *MAPK* signalling pathway-plant, plant–pathogenic agent mutual effect, and plant hormone signal transduction. After calculating the correlation of RNA sequencing and qRT-PCR outcomes in combination with the obtained expression profiles for *Frl* and MM samples, the pairwise coefficients of association (R^2^) ranged from 0.942 to 1.0 (Figure 7), indicating that the RNA-seq data were of high quality and could be utilised for subsequent analyses.

### 2.8. Determination of Critical Genes from Vital Pathways

The ‘*MAPK* signalling pathway-plant’ pathway is an important pathway involved in the mediation of plant resistance. As shown in Figure 8c, *MAPK* induces downstream *WRKY* genes. Hence, we analysed the expression levels of nine *MAPK* and twelve *WRKY* genes. The expression of eight *MAPK* genes was increased in *Frl* tomato compared with MM tomato at 3 dpi. Among these genes, the expression of four *MAPK* genes was higher in *Frl* than in MM at both 3 and 6 dpi (Figure 8a). Additionally, the expression of eight *WRKY* genes was higher in *Frl* tomato than MM tomato at 6 dpi, among which four *WRKY* genes showed higher expression in *Frl* than MM at both 3 and 6 dpi (Figure 8b).

### 2.9. Hormone Measurements

To investigate the hormone response to FORL infection, we measured JA and SA content at 0, 3, and 6 dpi. As shown in Figure 9a, the JA content continued to increase in *Frl* tomato after inoculation, whereas the JA content in MM tomato decreased gradually. Interestingly, the degree of change in both tomato cultivars was more significant at 0–3 dpi than 3–6 dpi. The SA content in both *Frl* and MM tomatoes peaked at 3 dpi (Figure 9b) and then quickly decreased between 3 and 6 dpi. Both JA and SA levels were generally higher in *Frl* than MM. These results indicated that JA and SA increased during the early stage of infection, but the change in JA was more obvious. Thus, these hormones are vital to the defence mechanisms in *Frl* tomato plants exposed to FORL.

### 2.10. Analysis of DEGs in Response to Biotic Stress

MapMan software was used for the visual analysis of DEGs at 3 dpi in these tomato cultivars (Figure 10). The outcomes showed that most enrichment of DEGs occurred in transcription factors, protein modifications, and protein decomposition. The DEGs mainly responded to JA and SA. The DEG expression trends in the samples at 3 dpi were similar, and there was more gene downregulation than gene upregulation. In the redox reaction, the DEGs were involved in biological regulation by receptor kinases, calcium regulation, G-proteins, light, MAP kinases, phosphoinositides, carbon and nutrition, and unspecified pathways. Moreover, some DEGs accumulated in several redox enzyme classifications, including ascorbate/glutathione, glutaredoxin, thioredoxin, dismutase/catalase, peroxiredoxin, and haem. The DEGs in the comparison group Frl3dpi vs. MM3dpi were chosen for biotic analyses, and the results showed that the host cells recognised and released *R* genes when pathogens attacked, triggered signalling and *MAPK* pathways, released transcription factors, and stimulated defence genes to release PR proteins to complete the defence response. In the biotic stress pathway, there were more upregulated genes in *Frl* tomato than in MM tomato at 0–3 dpi.

## 3. Discussion

Herein, our team completed the first systematic transcriptomic analysis of the response to FORL. Microscopy observation of the effects of pathogenic agent in *Frl* and MM plants showed that 3 dpi was an important point in time when necrotic lesions appeared in MM plants and HR occurred in *Frl* plants. Considering these characteristics, together with the observed changes in gene expression, the infection process can be divided into early and late stages [26]. Horizontal (comparisons of genetic expression in identical plants at diverse temporal points) and vertical (comparisons of gene expression in different plants at identical temporal points) contrasts were performed for the different cultivars and infection stages. The results indicated that among the DEGs identified in the horizontal comparisons, more DEGs were identified in disease-tolerant and disease-susceptible plants during the early stage (0–3 dpi) than the late stage (3–6 dpi). Among the DEGs identified in the vertical contrasts, fewer DEGs existed in the resistant plants than the susceptible plants during the early phase, whereas more DEGs were noted in the resistant plants than the susceptible plants during the late stage. In summary, the resistance gene-mediated defence response was activated during the early stage, and the gene expression level was relatively stable in the disease-tolerant plants compared to the disease-susceptible plants. These results are consistent with those reported for other genes mediating resistance in tomato [27,28] and other plants [29].

The *MAPK* pathway is an important pathway involved in plant stress responses [30,31,32,33] and consists of *MAPK*, mitogen-activated protein kinase (MAPKK), and mitogen-activated protein kinase kinase (MAPKKK) components (Figure 8c) [34,35]. The expression levels of 9 *MAPK* genes were analysed, and the outcomes revealed that the expression of genes with the following IDs was higher in *Frl* tomato than MM tomato at 3 dpi: 543859 (*MAPK10*), 100736548 (*MAPK14*), 100529134 (*MAPKKKe*), 100534642 (*MAPK5*), 100735510 (*MAPK8*), 100736539 (*MAPK16*), 543918 (*MAPKKKalpha*), and 100736475 (*MAPK12*). Notably, the expression of genes with the following IDs was higher in *Frl* tomato than MM tomato at both 3 and 6 dpi: 543859 (*MAPK10*), 100736548 (*MAPK14*), 100529134 (*MAPKKKe*), and 100534642 (*MAPK5*) (Figure 8a). In conclusion, *MAPK* genes were shown to participate in the plant disease resistance immuno-response mechanism and were relatively active during the early stage of infection. *MAPK* cascades accelerate the rapid death of plant cells at the site of infection by inducing the expression of defence genes and the accumulation of ROS, thereby inhibiting the further spread of the pathogen. This result coincides with the results of Yoshihiro et al. [36] and Lee et al. [37].

*WRKY* is a type of plant-specific TF vital for plant stress tolerance. Abiotic stress caused by the surrounding environment, including drought, soil salinity, low temperature, and heavy metal pollution, also rapidly increases the expression levels of *WRKY* transcription factors [38,39,40]. The results revealed that *WRKY* family members were involved in pathogen infection and pathogen attack via the *MAPK* pathway (Figure 8c). The expression levels of 12 *WRKY* genes were analysed, and the results revealed that the expression of the genes with IDs 100301944 (*WRKY40*), 101256570 (*WRKY80*), 101259967 (*WRKY81*), and 101268780 (*WRKY46*) was higher in *Frl* tomato than in MM tomato at 3 and 6 dpi. In addition, the expression of the genes with IDs 100125891 (*WRKYIId-1*), 101255501 (*WRKY31*), 100736444 (*WRKY41*), and 101247683 (*WRKY6*) was higher in *Frl* tomato than MM tomato at 6 dpi (Figure 8b). These results indicated that these *MAPK* genes might stimulate various *WRKY* genes downstream and thus be pivotal for the resistance reaction of *Frl* tomato to FORL, as shown in Figure 8c.

Plant hormones modulate the expression of genetic networks involved in defence reactions, among which JA and SA constitute the hormonal backbone of plant immunity [41]. The JA pathway in plants is mainly activated by insect injury and necrotic microorganism invasion, and SA mainly mediates resistance to biotrophic and semi-biotrophic pathogens [42]. Therefore, we can infer that SA is the main mediator during the early stage of the hormonal response to infection. Once necrotic spots started to appear on the plant stem, JA was the main endogenous hormone participating in the disease-tolerance reaction. Synergistic and antagonistic effects between SA and JA signalling pathways are considered to exert regulatory effects in plants for survival in complex biological environments [43].

## 4. Materials and Methods

### 4.1. Plant Materials and FORL Inoculation

Two tomato cultivars, the disease-resistant cultivar ‘19912’, harbouring the *Frl* gene, and the susceptible cultivar ‘Moneymaker’ (MM), which lacks FORL-resistance genes (provided by the Tomato Research Genetic Resource Center), were utilised herein. FORL was cultured in potato dextrose agar (PDA) liquid medium at 28 °C with shaking at 120 rpm for 4–5 days. At the four-to-six-leaf stage, we artificially injected FORL at a concentration of 1 × 10^7^ sporangia per millilitre [44] into the base of the tomato stem [45]. Samples were maintained in a greenhouse at 28 °C with greater than 90% relative humidity at our university (Northeast Agricultural University, Harbin, China). Tomato specimens were collected in triplicate before inoculation and at 3 and 6 days after inoculation (dpi). All specimens were rapidly placed in LN (liquid nitrogen) and preserved at −80 °C for RNA extraction.

### 4.2. Microscopic Observation of FORL in Frl Tomato

To identify the *Frl*-mediated hypersensitive response (HR) and the crucial temporal points regarding the cause of the disease tolerance response, we observed plant stems following trypan blue staining [46] and with scanning electron microscopy (SEM) [47]. Stem specimens were harvested prior to exposure to disease and at 3 and 6 dpi.

### 4.3. RNA Extraction and Transcriptome Sequencing

Total RNA was acquired from each group from a total of 18 samples (0, 3, and 6 dpi in triplicate for both cultivars) via the RNA Preparation Pure Plant tool (Thermo Fisher, Waltham, MA, USA) for qRT-PCR and RNA sequencing analyses [48]. RNA sequencing was completed by BGI Tech (Shenzhen, China) in a process that mainly included the following steps: mRNA was separated from the overall RNA via oligo (dT) and purified. The purified mRNA was utilised to construct 18 transcriptome libraries, which were then sequenced using the DNBseq machine-based probe-anchored polymerization technique [26,49].

### 4.4. RNA Sequence Read Matching and Differentially Expressed Gene (DEG) Identification

Clean reads in FASTQ format were obtained from SOAPnuke (v1.5.2) after filtration of the original data. The reads were filtered to remove reads with a low-quality base (base quality ≤ 5) frequency greater than 20%, reads with greater than 5% poly-N (unknown base) content, and sequencing adapters.

Clean reads were matched to the reference genomic set via Bowtie2 (v2.2.5) [50]. HISAT2 (v2.0.4) was utilised to map the clean data to the *Solanum lycopersicum* reference genome sequence (NCBI_GCF_000188115.4_SL3.0) [51]. Genetic expression levels were acquired based on the fragments per kilobase of transcript per million (FPKM) approach with RSEM (v1.2.12) [52]. The analysis of DEGs was conducted via DESeq2 (v1.4.5) according to a Q value ≤ 0.05 [53].

### 4.5. Functional Annotation and Enrichment Analysis of DEGs

To identify the roles of these DEGs, significantly enriched GO terms and KEGG pathways were analysed using Phyper (https://en.wikipedia.org/wiki/Hypergeometric_distribution (accessed on 1 April 2021)) on the foundation of the hypergeometry test. A *p* ≤ 0.05 was deemed highly significant for GO terms and KEGG pathways [54].

### 4.6. Determination of Endogenous Hormones

The SA and JA pathways are two important signalling pathways in plant-based resistance to gene-mediated resistance and induced resistance [55,56]. JA and SA were abstracted from leaves using the approaches of Llugany et al. [57] and Liu et al. [58] with minor modifications and measured by liquid chromatography–mass spectrometry (LC–MS).

### 4.7. Weighted Gene Co-Expression Network Analysis (WGCNA)

We performed WGCNA on the DEGs to determine co-expressed genetic modules and to investigate the relationship between genetic networks and phenotypes of interest along with the hub genes at the centre of the network [28].

### 4.8. MapMan Analysis of the Biofunctions of DEGs

MapMan is plant-specific, mostly manual, pathway analysis software that provides a good visual interface and can directly map gene expression data on the pathway map, providing complete gene functional classification and a comprehensive pathway map [59]. Based on the transcriptome data of tomato, a matching file was established using the website http://www.plabipd.de/portal/mercator-sequence-annotation (accessed on 1 April 2021).

### 4.9. qRT-PCR Analysis

Fourteen DEGs were screened, and their expression profiles, as determined by RNA-seq, were verified by qRT-PCR. The primers for these analyses were developed using Primer 5.0 software. The data were quantified via the 2^−ΔΔCT^ approach, and *EFα1* was used for normalisation (R: 5′-CCACCAATCTTGTACACATCC-3′, S: 5′-AGACCACCAAGTACTACTGCAC-3′) [60]. qRT-PCR was performed using a qTOWER3G Detection System (Analytik Jena, Jena, Germany) and AceQ qPCR SYBR Green Master Mix (Vazyme, Nanjing, China).

## 5. Conclusions

In summary, 189 DEGs in the ‘plant–pathogen interaction’ pathway were involved the response to FORL. Artificial inoculation and microscopy analyses revealed symptoms in the susceptible MM cultivar at the site of inoculation at 3 dpi. RNA-seq analyses were completed to determine the FORL-responsive regulation pathways. As the mycelium triggered an immune response and the secretion of effector proteins, several downstream pathways and associated defence signal transduction pathways were also triggered, including pathways involving MAPK, *WRKY*, JA, and SA. Among these components, *WRKY* proteins can activate R genes and regulate various downstream resistance genes. Eventually, HR was induced, triggering cellular death around the infected site and restricting pathogen growth. The expression levels of genes that were higher in 19912 than MM at both 3 and 6 dpi included *MAPK10*, *MAPK14*, *MAPKKKe*, *MAPK5*, *WRKY40*, *WRKY80*, *WRKY81*, and *WRKY46*. These outcomes reveal the underlying molecular mechanism opposing FORL infection in *Frl* tomato and lay a foundation for further screening of *Frl* genes.

## Figures and Tables

**Figure 1 ijms-23-07078-f001:**
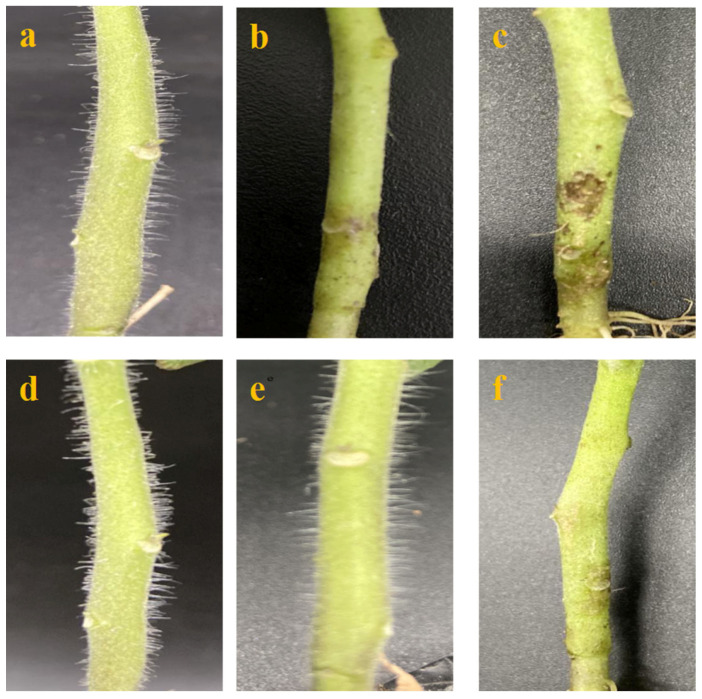
Results of inoculation. (**a**–**c**) ‘Moneymaker’ inoculated with FORL (*Fusarium oxysporum* f. sp. *radicis-lycopersici*) at 0, 3, and 6 dpi; (**d**–**f**) ‘19912’ inoculated with FORL at 0, 3, and 6 dpi.

**Figure 2 ijms-23-07078-f002:**
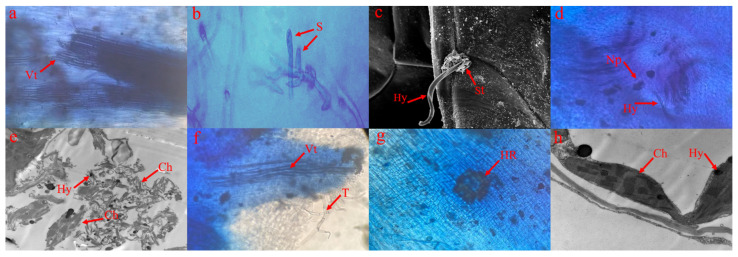
Trypan blue staining of tomato stem samples exposed to FORL. (**a**–**e**) ‘Moneymaker’ trypan blue staining at 0, 1, 2, 3, and 6 dpi; (**f**–**h**) ‘19912’ tomato stems dyed with trypan blue at 0, 3, and 6 dpi. Vt, vascular tissue; S, spores; Hy, hyphae; St, stomata; Np, necrotic plaque; T, trichome; HR, hypersensitive response; Ch, chloroplast.

**Figure 3 ijms-23-07078-f003:**
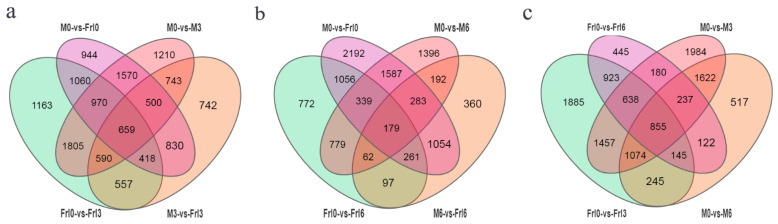
Venn diagrams presenting DEGs in diverse comparisons after FORL inoculation. (**a**) Venn chart of DEGs identified in the M0 vs. Frl0, M0 vs. M3, Frl0 vs. Frl3, and M3 vs. Frl3 comparisons. (**b**) Venn diagram of DEGs identified in the M0 vs. Frl0, M0 vs. M6, Frl0 vs. Frl6, and M6 vs. Frl6 comparisons. (**c**) Venn diagram of DEGs identified in the Frl0 vs. Frl6, M0 vs. M3, Frl0 vs. Frl3, and M0 vs. M6 comparisons.

**Figure 4 ijms-23-07078-f004:**
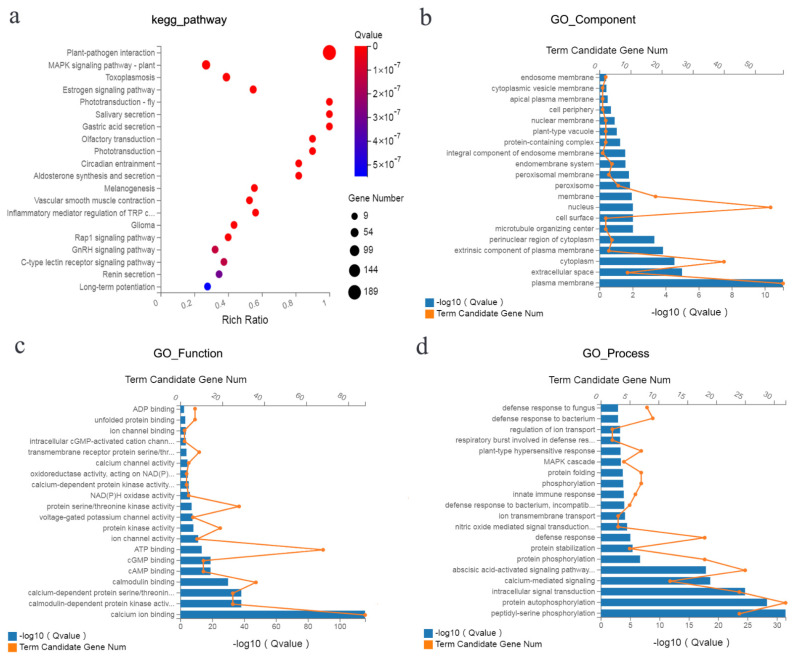
(**a**) KEGG enrichment of 189 DEGs in the ‘plant–pathogen interaction’ pathway. (**b**–**d**) GO enrichment of 189 DEGs in the ‘plant–pathogen interaction’ pathway.

**Figure 5 ijms-23-07078-f005:**
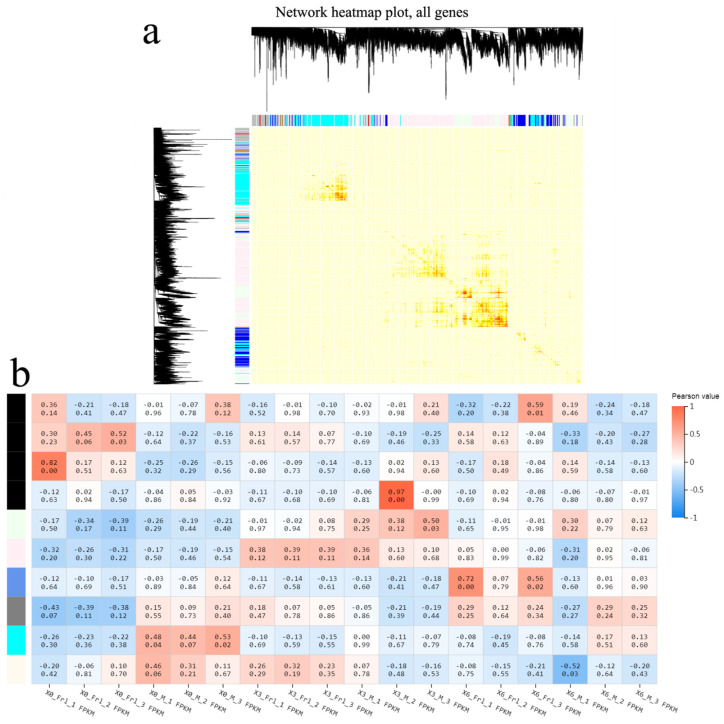
Gene co-expression network analyses via WGCNA. (**a**) Gene tree coloured per the relationship between genetic expression levels. Different colours denote different genetic modules and reflect different coefficients between genes. (**b**) Module–specimen relationship. The abscissa denotes the specimens, and the ordinate denotes the modules. The numbers in each cell denote the coefficients of association (**top**) and P results (**bottom**). The alteration from blue (low) to orange (high) denotes the DEG ranges.

**Figure 6 ijms-23-07078-f006:**
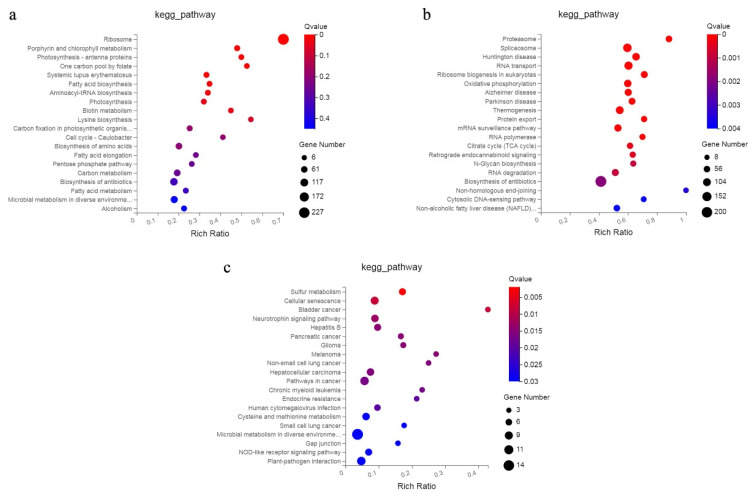
Genes in three modules were analysed using KEGG. (**a**–**c**) Results for the MElightcyan, MEpink, and MEblue modules, respectively.

**Figure 7 ijms-23-07078-f007:**
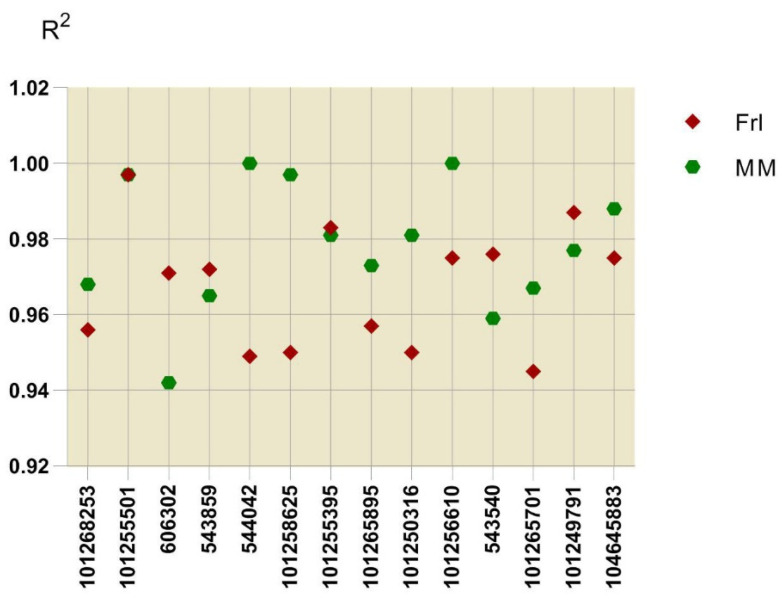
Coefficients of association between RNA sequencing and qRT-PCR results. The expression patterns of each gene in both *Frl* (0, 3, and 6 dpi) and MM (0, 3, and 6 dpi) were analysed. The outcomes acquired from these approaches were utilised to compute the coefficients of association (R^2^ values). All points denote R^2^ values. MM, ‘Moneymaker’; *Frl*, ‘19912’.

**Figure 8 ijms-23-07078-f008:**
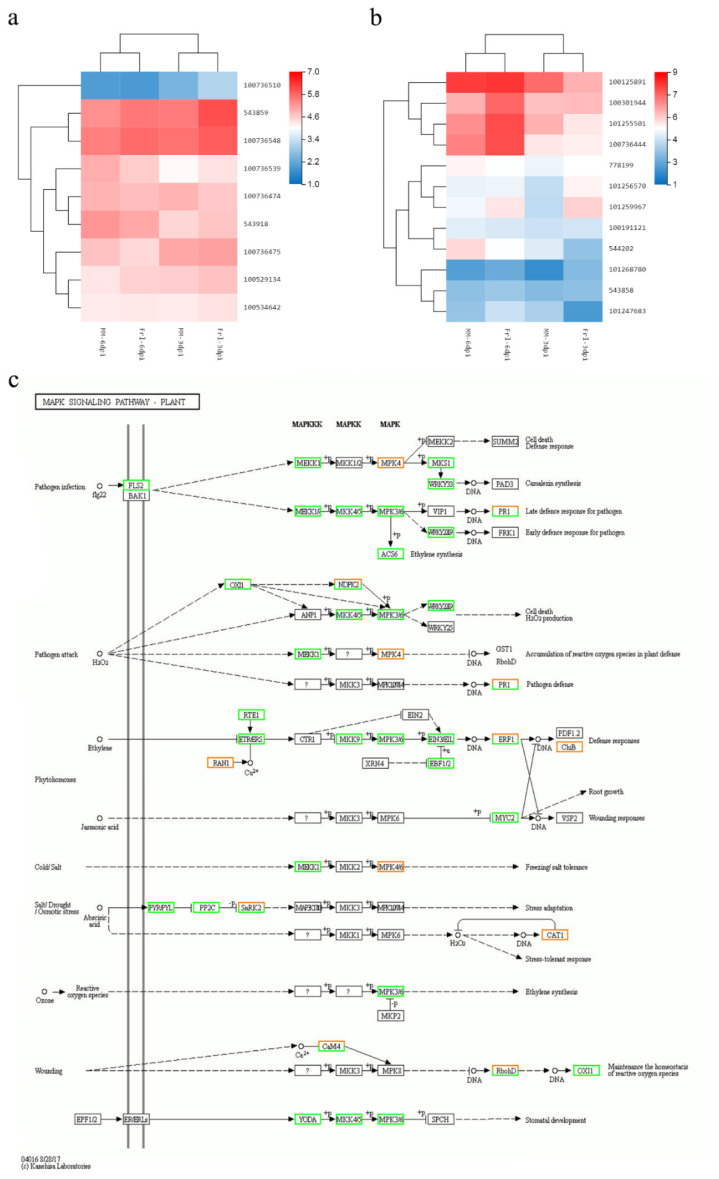
(**a**) DEGs related to *MAPK* in *Frl* and MM samples. (**b**) DEGs related to *WRKY* in *Frl* and MM samples. (**c**) Network analysis of ‘*MAPK* signalling pathway-plant’ predicting the response to FORL infection.

**Figure 9 ijms-23-07078-f009:**
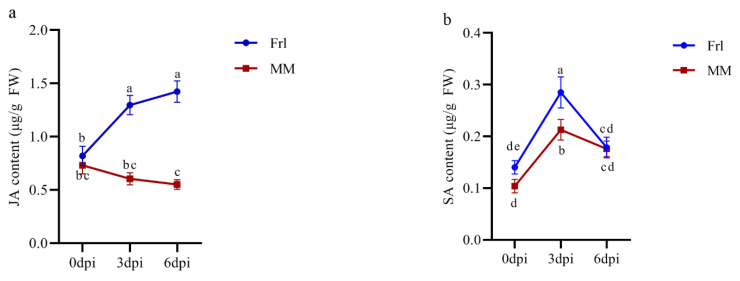
Fluctuations in (**a**) JA and (**b**) SA as a function of days after FORL infection in *Frl* tomato and MM tomato. JA, jasmonic acid; SA, salicylic acid; *Frl*, ‘19912’; MM, ‘Moneymaker’; dpi, days post-infection. Different letters indicate siginificant differences.

**Figure 10 ijms-23-07078-f010:**
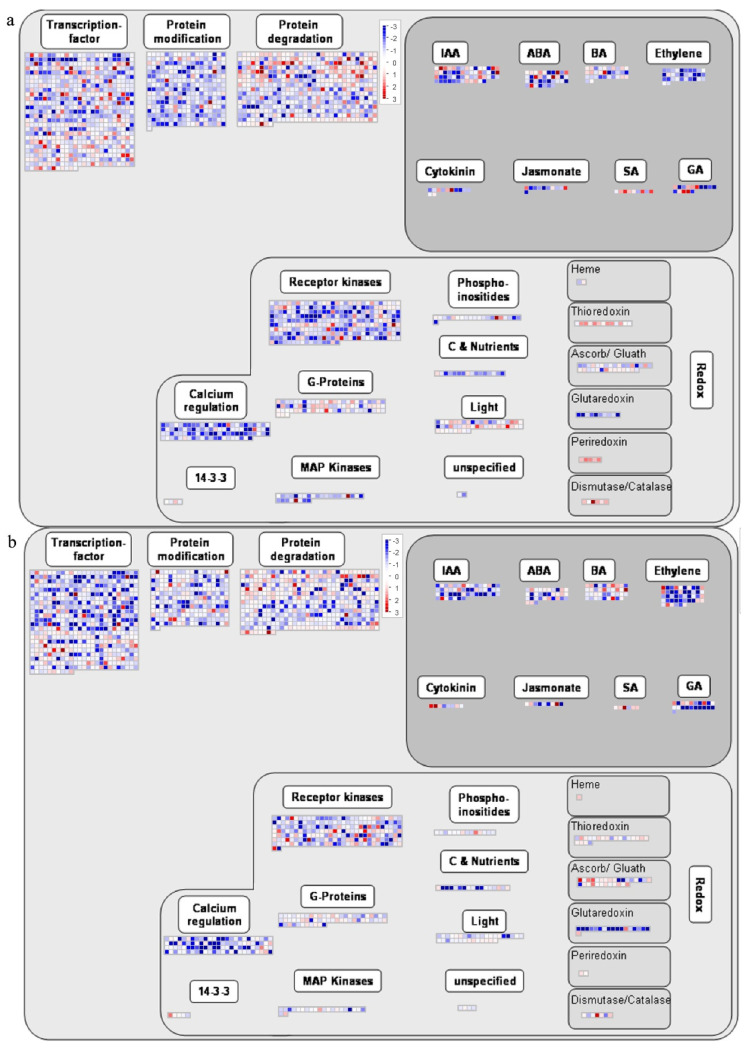
Cluster pattern of DEGs produced via MapMan. (**a**) DEG modulation overview for *Frl* tomato at 0–3 dpi. (**b**) DEG modulation overview for MM tomato at 0–3 dpi. (**c**) Biotic stress and DEG regulation in *Frl* tomato vs. MM tomato at 3 dpi. (**d**) Cell reaction and DEG regulation in *Frl* tomato at 0–3 dpi. (**e**) Cell reaction and DEG regulation in MM tomato at 0–3 dpi.

**Table 1 ijms-23-07078-t001:** Statistics for DEGs exhibiting different expression patterns.

DEG Set	Total DEGs	Upregulated	Downregulated
Frl0 vs. Frl3	7222	3374	3848
Frl0 vs. Frl6	3545	1567	1978
Frl3 vs. Frl6	4291	2398	1893
M0 vs. M3	8047	3972	4075
M0 vs. M6	4817	2316	2501
M3 vs. M6	2944	1473	1471
Frl0 vs. M0	6951	3487	3464
Frl3 vs. M3	5039	2358	2681
Frl6 vs. M6	2488	1222	1266

0 vs. 3, comparison between 0 and 3 dpi; 3 vs. 6, comparison between 3 and 6 dpi; 0 vs. 6, comparison between 0 and 6 dpi; dpi, days post-inoculation.

## Data Availability

All of the data pertaining to the present study have been included in the tables and figures of the manuscript. The raw sequencing data were deposited in NCBI GEO under accession number GSE171219 (https://www.ncbi.nlm.nih.gov/geo/query/acc.cgi?acc=GSE171219, accessed on 25 May 2022).

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
