# Peer review of "Transcriptome Analysis Reveals the Response Mechanism of Frl-Mediated Resistance to Fusarium oxysporum f. sp. radicis-lycopersici (FORL) Infection in Tomato"

_ijms, 2022, doi:10.3390/ijms23137078_

Round 1

Reviewer 1 Report

The presented work on fusarium crown and root rot (FCRR) is highly significant for peer tomato researchers to advance knowledge about genetic resistance against FCRR. The authors have presented their work succinctly; however, the presented research is not strong enough to be considered in the current form hence recommend to make a major revision for further consideration.

Also, the use of English seems to be inappropriate in places and it is recommended that authors review the manuscript from a native English speaker to correct grammatically incorrect sentences. There are some minor issues reported in regard to typos, please see the review comments given below.

NOTE: I wish the authors provided the line numbers so that specific review comments could have been provided for the relevant review comments.

Author Response

Response to Reviewer 1 Comments

The manuscript has been linguistically edited.

Point 1: Abstract: Abstract is written well and has rendered general significance and synopsis of the presented research, which could be useful for broader readership. However, current abstract is longer than the recommended word count (200 words) hence it is recommended that authors further trim the abstract and present is succinctly. Authors could avoid using abbreviations which are commonly used by peers such KEGG, JA, SA, and qRT-PCR. Also, authors have missed providing research hypothesis, which is important part of the research goal and would encourage authors to revise the abstract to consider the made suggestions.

Response 1:  The abstract has been revised to 199 words according to your comments.

Point 2: Introduction: Introduction is succinct, but still could be improved to make it more informative.

Also, some lines are disconnected for reason (For example “ is activated by SA” has a gap between the words and it mimics as if there is another paragraph).

Response 2: This section was modified according your kind advice. (Line 58)

Point 3: Results: 2.1. Inoculation: “MM” Please provide the full form on first appearance before providing

abbreviation.

Response 3: “MM” has been provided the full form on first appearance. (Line 79)

Point 4: 2.2. Microscopic analysis of FORL growth in two tomato cultivars: “Figure 2” Please provide

better image, as current image is very hard to interpret.

Response 4: A clearer picture has been uploaded for Figure 2.

Point 5: 2.3. RNA sequencing and identification of transcripts: “Supplementary Information is missing

the submitted manuscript” hence cannot verify the presented information.

Response 5: Supplementary Information has been uploaded.

Point 6: 2.4. DEGs in response to FORL inoculation: “Table 1” what is EDG? Seems likely typo for

DEG Table 1. Total DEGs Sum “4237” for Frl2 vs. Frl6 set is incorrect as it is “4291” Figure 3. Figure captions are too small and are unreadable.

Response 6: ‘EDG’ has been modified to ‘DEG’; ‘4237’ has been modified to ‘4291’; A clearer picture has been uploaded for Figure3.

Point 7: 2.5. KEGG pathway and GO enrichment analysis of DEGs: “Figure 4 is unreadable and hard

to understand, please provide the alternative figure with better resolution and size”. Authors have

provided no rationale of presenting this figure as there is hardly any explaination for each subfigure

“Figure 4a-4d”. There is a lot of information in this figure and can assist readers to comprehend to understand the relationship between different components.

Response 7: The alternative figure with better resolution and size has been uploaded for Figure 4.

Point 8: 2.6. Gene co-expression network analysis: Do not start the paragraph with the abbreviation

“Figure 5a seems to provide no information so not able to understand the rational of providing the

same”

Response 8: The abbreviation has been modified; Figure 5a showed genetic tree coloured per the relationship between genetic expression levels. Diverse colours denote diverse genetic modules and reflect different coefficients between genes. Due to the large number of DEGs, the color partitions were not very obvious.

Point 9: Discussion: The result findings are not robustly discussed as it sounds more of explanation about

MPAK and WRKY genes rather than rigorous discussion on how the observed results of both genes in JA and SA pathways would impact on overall defense mechanism and what it means for their respective role during FCRR infection and developing potential resistance.

Response 9: Thank you very much for your kind advice. How MAPK and WRKY gene families interact with JA and SA pathways to affect the overall defense mechanism will be the direction we need to further study in the future. Your suggestions are very constructive. We will dig into it in the follow-up research.

Point 10 :Materials and Methods: Sufficient methodology details are provided and peer researchers can

repeat similar experiment if needed.

Response 10: References have been cited for the experimental methods of these five parts (4.1; 4.2; 4.3; 4.6; 4.9), and the methods of these four parts (4.4; 4.5; 4.7; 4.8) are made with the help of corresponding tools, which have been listed in the paper. If you need any other methods, please tell me, and I will try to modify this manuscript.

Poimt 11: Conclusion: Few conclusions are provided but still key take home messages are missing.

Response 11: Some new content has been added to the conclusion (Line 678)

Point 12: References: Around 10% (5, 16, 26, 33, 43, 48) cited publications are self-citation so authors are

encouraged providing self-citations where they are highly relevant.

Response 12: References 5, 16, 26, 33, 43, 48 have been replaced with other references. (Line 714, 742, 769, 787, 812, 824)

Finally, the manuscript has been revised according to comments of reviewers and editor. If there are any shortage, please inform us. We will try our best to revise this manuscript.

Thank you very much for your attention and kind advice.

Your Sincerely,

Dr. Yuqing Sun

Reviewer 2 Report

This paper reports original data on gene transcription after infection with a key pathogen of tomato in two cultivars with different levels of susceptibility. Undoubtedly, these results are relevant for a better comprehension and management of crown-root rot of tomato, and I believe that they deserve to be divulged to the scientific community. The manuscript is well written, and only requires minor revision for the following points:

Key words: 'Molecular mechanism' is too generic; choose another one. Introduction: to the best of my knowledge, tomato yellow leaf curl virus is spread by Bemisia tabaci; page 2, line 6: do not start a new line. Results: meaning of abbreviations such as MM, dpi, HR, DEG must be specified on first mention in the text; in this respect, manuscript should be checked throughout. Section 4.2: Fusaria do not produce sporangia: do you mean conidia?

Author Response

Response to Reviewer 2 Comments

Point 1: Key words: 'Molecular mechanism' is too generic; choose another one. 

Response 1: Key words: 'Molecular mechanism' has been replaced with ‘resistance response’. (Line 22)

Point 2: Introduction: to the best of my knowledge, tomato yellow leaf curl virus is spread by Bemisia tabaci; 

Response 2: You are right. Tomato Fusarium crown and root rot is a soil-borne disease.

Point 3: page 2, line 6: do not start a new line. 

Response 3: Format error, corrected. (Line 58)

Point 4: Results: meaning of abbreviations such as MM, dpi, HR, DEG must be specified on first mention in the text; in this respect, manuscript should be checked throughout. 

Response 4: Meaning of abbreviations such as MM, dpi, HR, DEG have been specified on first mention in the text. (Line 82, 86)

Point 5: Section 4.2: Fusaria do not produce sporangia: do you mean conidia?

Response 5: I only found sporangia in section 4.1, which refers to conidia. (Line 331)

Finally, the manuscript has been revised according to comments of reviewers and editor. If there are any shortage, please inform us. We will try our best to revise this manuscript.

Thank you very much for your attention and kind advice.

Your Sincerely,

Dr. Yuqing Sun

Round 2

Reviewer 1 Report

Authors have made signicant corrections and have revised the manuscript as per the provided review comments. Revised manuscript reads well now.